# Impact of Wuhan lockdown on the indications of cesarean delivery and newborn weights during the epidemic period of COVID-19

Min Li[1☯], Heng Yin[1☯], Zhichun Jin[2], Huan Zhang[1], Bingjie Leng[1], Yan Luo[1], Yun Zhao[1]*

1 Department of Obstetrics, Maternal and Child Health Hospital of Hubei Province Tongji Medical College, Huazhong University of Science and Technology, Wuhan, China, 2 Department of Traditional Chinese Medicine, Maternal and Child Health Hospital of Hubei Province Tongji Medical College, Huazhong University of Science and Technology, Wuhan, China

☯ These authors contributed equally to this work.

* zhao020060@163.com

**Data Availability Statement:** All relevant data are within the manuscript and its Supporting Information files.

## Abstract

### Objective

To prevent the rapid spread of COVID-19, the Chinese government implemented a strict lockdown in Wuhan starting on 23 January, 2020, which inevitably led to the changes in indications for the mode of delivery. In this retrospective study, we present the changes in the indications for cesarean delivery (CD) and the birth weights of newborns after the lockdown in Wuhan.

### Methods

A total of 3,432 pregnant women in the third trimester of their pregnancies who gave birth in our hospital from 23 January 2019 to 24 March 2020 were selected as the observation group, while 7,159 pregnant women who gave birth from 1 January 2019 to 22 January 2020 were selected as the control group; control group was matched using propensity score matching (PSM). A comparative analysis of the two groups was performed with the chi-square test, $t$ test and rank sum test.

### Results

The difference in the overall rate of CD between the two groups was not statistically significant ($p < 0.05$). Among the indications for CD, CD on maternal request (CDMR) and fetal distress were also significantly more common in the observation group ($p < 0.05$) than the control group. Furthermore, we found that the weight of newborns was significantly heavier in the observation group than in the control group when considering full-term or close-to-full-term births ($p < 0.05$).

### Conclusions

The results may provide useful information to management practices regarding pregnancy and childbirth after lockdown in other cities or countries, enabling better control of the rate of

**Funding:** The authors received no specific funding for this work.

**Competing interests:** The authors have declared that no competing interests exist.

CD due to CDMR, reducing fetal distress, and controlling newborn weight. We recommend that pregnant women pay more attention to controlling the weight of newborns through diet and exercise.

## Introduction

At the end of December 2019, a cluster of cases of coronavirus disease-2019 (COVID-19) were first reported in Wuhan, China [1]. On 23 January 2020, the municipal government of Wuhan announced the lockdown of the entire city, and China implemented a national emergency mechanism with different cities adopting different measures according to their respective situations. On 30 January 2020, the WHO declared COVID-19 to be a "public health emergency of international concern", and the US recorded its first confirmed case of human-to-human transmission in Chicago [2, 3]. It is evident that the lockdown in Wuhan has played a critical role in limiting the scope of the COVID-19 epidemic in China.

Since then, the life-threatening COVID-19 outbreak has become a global pandemic. By 5 April 2020, the global number of confirmed cases had exceeded 1.1 million, with a daily increase of approximately 0.1 million cases. To cope with this disaster, an increasing number of government agencies in the US and European countries have implemented lockdown policies to prevent the rapid spread of COVID-19. During the lockdown, city residents are required to refrain from leaving home and to practice social distancing. In the past two months, scholars began to pay attention to the management of pregnancy and childbirth during the COVID-19 pandemic. Previous studies have shown that pregnancy while infected with severe acute respiratory syndrome coronavirus (SARS-CoV) was associated with adverse maternal and neonatal complications, such as spontaneous miscarriage, preterm delivery, intrauterine growth restriction, the need for endotracheal intubation, admission to the intensive care unit, renal failure, and disseminated intravascular coagulopathy [4, 5].

Although the possible risks associated with COVID-19 and the clinical characteristics of pregnant women with laboratory-confirmed COVID-19 have been investigated [6, 7], previous research has failed to address the impact of lockdown during the COVID-19 epidemic. Unlike other activities that can be carried out online or at home, delivery cannot be delayed arbitrarily or performed at home. The lockdown in Wuhan may have a significant impact on the outcome of pregnancy due to pathological factors, the timeliness of examinations, psychological panic, limited medical resources, difficulty accessing and transportation, etc. An essential question that is important for obstetricians to consider is how does the lockdown affect the indications for different modes of delivery and newborn health?

This study answers the question based on an analysis of the data from more than 10,000 cases from our working hospital in Wuhan, the Maternal and Child Hospital of Hubei Province, which has one of the largest obstetrics departments of all Chinese hospitals. The department delivered approximately 25,000 babies annually in the last five years. During the COVID-19 epidemic, our hospital was identified as a non-designated hospital with a fever clinic accept pregnant women who were not infected with severe acute respiratory syndrome coronavirus 2 (SARS-CoV-2) and those who were suspected of being infected. The latter patients who received laboratory-confirmation of infection via quantitative qPT-PCR for SARS-CoV-2 using a throat swab were transferred to designated hospitals for further treatment. During the lockdown in Wuhan, a total of 3,432 pregnant women underwent delivery in our birth center, and a few cases of COVID-19 were confirmed in pregnant women. In this

study, we compared the indications for cesarean delivery (CD) and the birth weights of newborns born to 3,432 pregnant women who gave birth during lockdown and 7,159 propensity score-matched pregnant women who gave birth before the lockdown were selected as the control group.

## Methods

### Study design and patients

Fig 1 illustrates the subject selection process. Pregnant women who had given birth in the birth center of the Maternal and Child Health Hospital of Hubei Province from 23 January 2020 to 14 March 2020 were enrolled as the observation group. The age of the selected patients ranged from 18 to 50 years old. The exclusion criteria included a gestational age less than 28 weeks and intrauterine fetal death. A total of 3442 pregnant women were initially included, among whom 5 with gestational age less than 28 weeks of and 5 with intrauterine fetal death were excluded. The remaining 3432 patients constituted the observation group.

From 29, 799 historical patients from 1 January 2019 to 22 January 2020, the control group included a total of 7,159 patients based on the propensity score matching (PSM) method as. The PSM method, a quasi-experimental design that has been used across disciplines to isolate treatment effects on a number of outcomes using observational data, was employed in this study to obtain matched patients to facilitate comparisons [8]. Based on three variables of age, gravidity, and parity, 7159 pregnant women were selected as the control group from the 29799 patients before the lockdown using the PSM method. The balanced scores of the two groups were 0.10312 and 0.10590, which were very similar.

### Data collection

In our department, one nurse is responsible for the daily recording of the delivery information of all the pregnant women into the Electronic Medical Record System (EMRS). In this research, obstetricians extracted the required data, including epidemiological, demographic, clinical, laboratory, and pregnancy outcome data for both the observation group and control group. The neonates' data including birth weight, clinical symptoms, Apgar score, and outcomes were also collected. During this period, all pregnant women with COVID-19 symptoms, such as fever, cough, and abnormal CT scan results, underwent nucleic acid detection of COVID-19 of SARS-COV02 from swab samples. In total, 13 pregnant women had confirmed COVID-19, including 8 pregnant women who underwent CD. Their throat swab samples were collected and sent to the local Chinese Center for Disease Control and Prevention (CDC), which was in charge of detecting SAS-COV-2 by qRT-PCR. The results for each woman suspected of having COVID-19 was entered into the EMRS.

### Statistical methods

Statistical analysis was performed using SPSS 22.0. The chi-square test was used for the comparison of countable data. The *t*-test and Fisher's exact probability method were used for the comparison of the means between the measurement data. The rank sum test was used for ranking variables. A difference was statistically significant at $p<0.05$.

## Results

A comparison of the observation group and control group is shown in Table 1. There were no significant differences in maternal age, gravidity, parity history, or BMI between the two

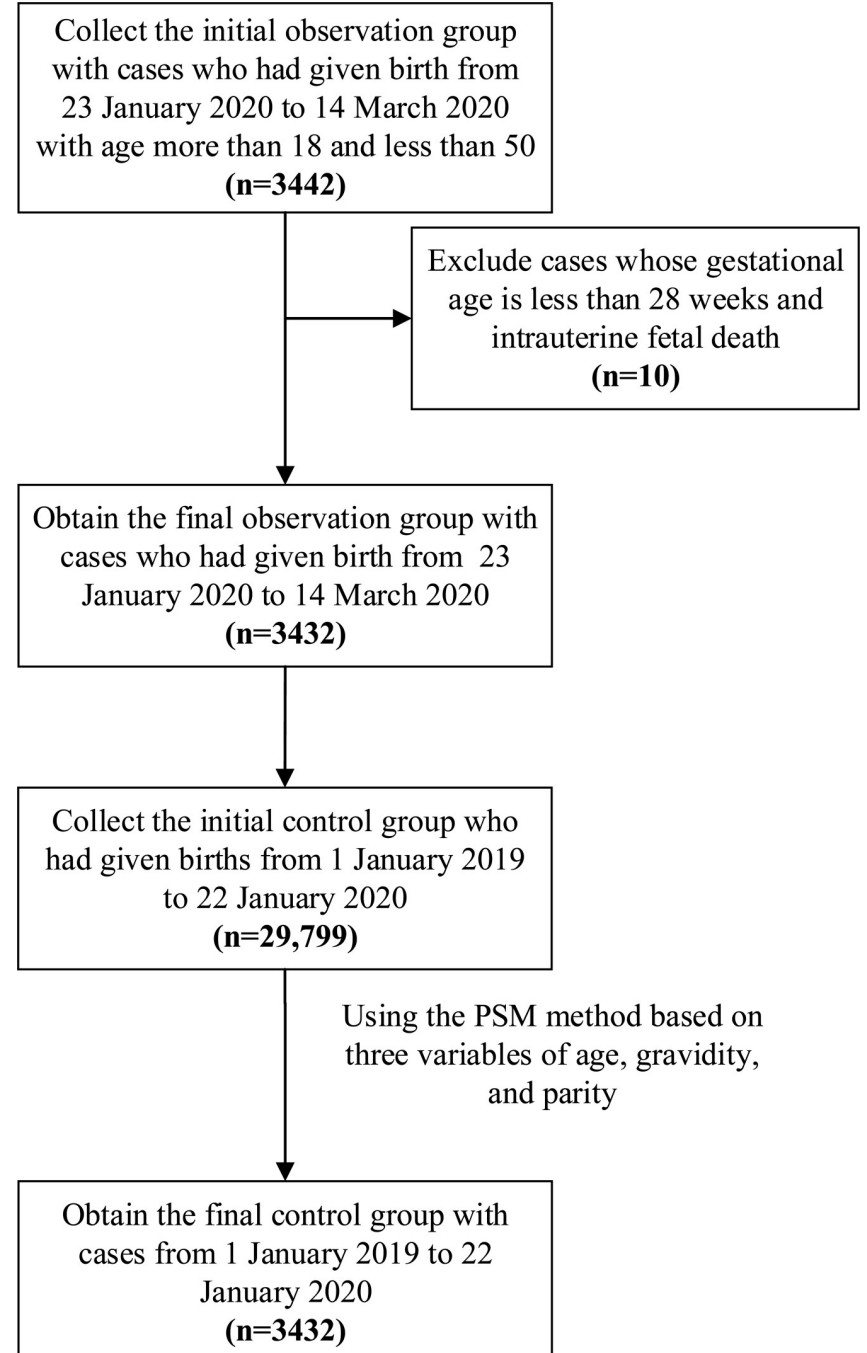

**Fig 1. Population flow chart of the retrospective study.**

groups ($p>0.05$). At the same time, there were no differences in the proportions of preterm and term neonates between the two groups ($p>0.05$).

Table 2 shows the comparison of the rates of CD and vaginal delivery before and after lockdown. The data show that there was no significant difference between the two groups ($p>0.05$). In fact, the CD rate in the observation and control groups were 47.49% and 47.70%, respectively.

**Table 1. Comparison of the general situation of pregnant women.**

| | | Before lockdown (n = 7159) | | After lockdown (n = 3432) | | Chi-square | *p*-value |
|---|---|---|---|---|---|---|---|
| | | No. | % | No. | % | | |
| Age | <35 | 6385 | 89 | 3095 | 90 | 2.43 | 0.12 |
| | ≥35 | 774 | 11 | 337 | 10 | | |
| Gravida | 1 | 3288 | 46 | 1544 | 45 | 1.44 | 0.70 |
| | 2 | 2003 | 28 | 981 | 29 | | |
| | 3 | 1113 | 16 | 527 | 15 | | |
| | ≥4 | 754 | 10 | 380 | 11 | | |
| Para | Primipara | 4685 | 65 | 2243 | 65 | 0.01 | 0.93 |
| | Multiparas | 2474 | 35 | 1189 | 35 | | |
| BMI | <25 | 6665 | 93 | 3167 | 92 | 2.35 | 0.16 |
| | ≥25 | 494 | 7 | 265 | 8 | | |
| Preterm and term births | Preterm | 615 | 9 | 281 | 8 | 0.49 | 0.47 |
| | Term | 6544 | 91 | 3151 | 92 | | |

Table 3 shows the comparison of the indications for CD. There were no significant differences in most of the pregnancy complications between the observation group and control group. The number of pregnant women who underwent CD due to fetal distress in the observation group was significantly higher than that in the control group ($p<0.05$). In addition, the incidence of CDMR in the observation group was significantly higher than that in the control group ($p<0.05$). CDMR was defined as a primary prelabor cesarean delivery performed on maternal request in the absence of any maternal or fetal indications [9]. In the observation group, 13 cases of COVID-19 were confirmed by chest CT scan and two positive laboratory tests for SARS-CoV-2 in throat swab samples. Eight of those 13 patients with confirmed COVID-19 underwent CD. There were 0 COVID-19 cases in the control group. The other indications for CD in the observation group were not significantly different than those in the control group.

Neonatal asphyxia is divided into mild asphyxia and severe asphyxia, which are mainly evaluated based on the Apgar score. According to the Apgar score, a total score from 0 to 3 is classified as severe asphyxia, from 4 to 7 is classified as mildly asphyxiated, and from 8 to 10 is normal [10]. Table 4 shows that there was no significant difference in neonatal asphyxia between the two groups ($p>0.05$).

As shown in Table 5, the neonatal birth weight in the observation group was heavier than that in the control group among those with ≥34 gestational weeks ($p<0.05$). However, there was no significant difference among those with fewer than 34 gestational weeks ($p>0.05$).

## Discussion

This paper presents a comparison of the indications for CD and newborn weights before and after the Wuhan lockdown. All the patients in the observation group and the control group

**Table 2. CD rate and vaginal delivery rate before and after lockdown.**

| | Before lockdown (n = 7159) | | After lockdown (n = 3432) | | Chi-square | *p*-value |
|---|---|---|---|---|---|---|
| | No. | % | No. | % | | |
| Vaginal delivery | 3759 | 53 | 1795 | 52 | 0.04 | 0.84 |
| Cesarean delivery | 3400 | 47 | 1637 | 48 | | |

**Table 3. Comparison of the indications of CD.**

| | Before lockdown (n = 3400) | | After lockdown (n = 1637) | | Chi-square | p-value |
|---|---|---|---|---|---|---|
| | No. | % | No. | % | | |
| Scar uterus | 1128 | 33 | 553 | 34 | 0.18 | 0.67 |
| Fetal distress | 537 | 15 | 297 | 18 | 4.41 | **<0.05** |
| Abnormal fetal position | 338 | 10 | 137 | 8 | 3.20 | 0.07 |
| Cesarean delivery on maternal request (CDMR) | 284 | 8 | 186 | 11 | 11.83 | **<0.05** |
| Giant fetus | 210 | 6 | 83 | 5 | 2.47 | 0.12 |
| Hypertension | 205 | 6 | 79 | 5 | 3.01 | 0.08 |
| Multiple pregnancy | 133 | 4 | 68 | 4 | 0.17 | 0.68 |
| Placenta previa | 94 | 3 | 32 | 2 | 2.97 | 0.09 |
| Induction of labor failure | 74 | 2 | 30 | 2 | 0.65 | 0.42 |
| Prenatal fever | 69 | 2 | 26 | 2 | 1.16 | 0.28 |
| Labor abnormalities | 46 | 1 | 22 | 1 | 0.02 | 0.98 |
| ICP[b] | 37 | 1 | 17 | 1 | 0.03 | 0.87 |
| Placental abruption | 33 | 1 | 13 | 1 | 0.38 | 0.54 |
| Diabetes | 31 | 1 | 12 | 1 | 0.42 | 0.52 |
| Umbilical cord | 22 | 1 | 5 | 0 | 2.42 | 0.12 |
| Genital malformation | 14 | 0 | 13 | 1 | 3.03 | 0.08 |
| FGR[c] | 11 | 0 | 1 | 0 | 3.20 | 0.07 |
| COVID-19 | 0 | 0 | 8 | 0 | | **<0.05**△[a] |
| Others | 34 | 1 | 17 | 1 | 0.02 | 0.90 |

[a]△means Fisher's exact probability method, because the frequency of cells appears 0

[b]ICP means intrahepatic cholestasis of pregnancy

[c]FGR means fetal growth restriction

were collected from the Maternal and Child Health Hospital of Hubei Province. During the lockdown to prevent the spread of the COVID-19 outbreak from 23 January 2020 to 13 March 2020, a total of 49,995 cases were confirmed in Wuhan with 2,446 deaths [11].

In our empirical study, the overall CD rate was not significant different between the observation group and the control group. The differences in most of the indications for CD between the observation group and control group were not significant. This may serve as reassurance to many pregnant women who are currently having to remain primarily at home that a lockdown is unlikely to dramatically affect complications of pregnancy. However, we empirically observed that the rate of CDMR increased significantly during this period. CDMR may reduce the risk of hemorrhage and transfusion, but is also potentially associated with a longer maternal hospital stay, an increased risk of respiratory problems for the infant, and the need for hysterectomy [9]. The main reason for the increase in the rate of CDMR was that the pregnant women in Wuhan were reluctant to wait for a natural birth due to fear of COVID-19 infection during hospitalization. After the city was locked down, 8 out of 13 pregnant women with

**Table 4. Comparison result on neonatal asphyxia.**

| | Neonatal asphyxia | | | | | | z-value | p-value |
|---|---|---|---|---|---|---|---|---|
| | Normal | | Mild | | Severe | | | |
| | No. | % | No. | % | No. | % | | |
| Before lockdown | 7097 | 99 | 51 | 1 | 11 | 0 | 0.54 | 0.46 |
| After lockdown | 3407 | 99 | 19 | 1 | 6 | 0 | | |

**Table 5. Newborn weight at different gestational age.**

| Newborn group | Before lockdown (g) | After lockdown (g) | t-value | p-value |
|---|---|---|---|---|
| 28 ≤ gestational age(wks) <32 | 1,462±318 | 1479±231 | -0.26 | 0.80 |
| 32 ≤ gestational age(wks) < 34 | 1,926±313 | 2,021±253 | -1.54 | 0.13 |
| 34 ≤ gestational age(wks) < 37 | 2,583±424 | 2,652±3856 | -2.06 | <**0.05** |
| 37 ≤ gestational age(wks) < 38 | 3,167±385 | 3,207±398 | -2.95 | <**0.05** |
| gestational age(wks) ≥39 | 3,402±370 | 3,426±384 | -2.32 | <**0.05** |

confirmed COVID-19 underwent CD in our birth center. Severe COVID-19 was treated as an indication for CD according to the Chinese Expert Consensus [12]. If a pregnant woman had mild or asymptomatic COVID-19 and her cervix was dilated, vaginal delivery could be selected. If the fetus was in distress, the level of control of COVID-19 was unsatisfactory, or there were other indications for CD, CD was performed by obstetricians [13].

During the Wuhan lockdown, pregnant women had to remain in their homes, limiting their ability to exercise and attend appointments. They reduced the frequency of prenatal examinations because they were afraid of contracting COVID-19, and some pregnant women were never examined from the initiation of lockdown until delivery. Therefore, some important risk factors may not have been detected in a timely manner, leading to an increased incidence of fetal distress.

Through the comparison of the observation group and the control group, we found that lockdown may have led to heavier newborns with a gestational age greater than 34 weeks. The possible underlying contributory factors might include food and nutrition changes and lack of exercise. During pregnancy, balanced nutrition and adequate intake of vegetables and protein have a positive impact on the birth weight of the neonate [14–16]. However, due to the COVID-19 in Wuhan, supermarkets and vegetable markets were closed, and access to nutrient-rich food was relatively restricted. The relative lack of vegetables and food high in crude fiber, and the increased intake of carbohydrate-rich foods, such as rice and noodles, which were easily obtained and stored, led to an increase in neonatal weight. In addition, after the city was locked down, pregnant women became less active and exercised less. We recommend that during a city lockdown due to COVID-19, pregnant women should try to eat a balanced diet, increasing their intake of protein and vegetables as much as possible and controlling their intake of carbohydrates such as rice noodles, and increase their engagement in indoor activities.

Our research has some limitations. First, due to the reduction in or even absence of examinations of pregnant women, we could not collect the data on changes in body weight during this period. Second, the findings might have been affected by other factors, such as climate, food type, and other uncontrollable inputs, because the observation group was collected from a short period in 2020. The research question investigated in this study needs further exploration with other data sets.

## Supporting information

**S1 Dataset.**
(XLSX)

## Author Contributions

**Conceptualization:** Min Li, Heng Yin, Yun Zhao.

**Data curation:** Min Li, Huan Zhang, Bingjie Leng, Yan Luo.

**Formal analysis:** Min Li, Yun Zhao.

**Investigation:** Zhichun Jin.

**Methodology:** Min Li.

**Project administration:** Yun Zhao.

**Resources:** Zhichun Jin.

**Supervision:** Yun Zhao.

**Validation:** Min Li, Heng Yin, Yun Zhao.

**Writing – original draft:** Min Li, Yun Zhao.

**Writing – review & editing:** Min Li, Yun Zhao.

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
