## [Decision Letter · Decision Letter 0]

1 Jun 2020

PONE-D-20-12899

The Influence of lockdown on the indications of cesarean delivery and newborn weights during the epidemic period of COVID-19 in Wuhan

PLOS ONE

Dear Dr. Zhao,

Thank you for submitting your manuscript to PLOS ONE. After careful consideration, we feel that it has merit but does not fully meet PLOS ONE’s publication criteria as it currently stands. Therefore, we invite you to submit a revised version of the manuscript that addresses the points raised during the review process.

This paper represents a resubmission of a previously submitted paper that was reviewed extensively. The authors were provided the opportunity to tone down their claims and ensure that throughout the paper it was clear that they are not examining how COVID-19 affects pregnancy outcomes or analyze such samples. They chose the former approach. The original reviewers have reviewed the revised paper and expressed conflicting recommendations. However, both agree that the work is not "blockbuster", but it may provide some comfort to pregnant women having to quarantine that it will not drastically affect their neonates. In reviewing the journal guidelines, it will allow for descriptive papers, such as this one. That being sad, it still needs to meet certain standards in terms of the English and previous and current reviewer concerns need to be addressed. It is recommended to have the revised manuscript proofed by an outside individual before it is resubmitted. Both reviewers have also provided helpful comments.

We look forward to receiving your revised manuscript.

Kind regards,

Cheryl S. Rosenfeld, DVM, PhD

Academic Editor

PLOS ONE

Journal Requirements:

Reviewers' comments:

Reviewer's Responses to Questions

**Comments to the Author**

1. Is the manuscript technically sound, and do the data support the conclusions?

Reviewer #1: Yes

Reviewer #2: Yes

2. Has the statistical analysis been performed appropriately and rigorously? 

Reviewer #1: Yes

Reviewer #2: Yes

3. Have the authors made all data underlying the findings in their manuscript fully available?

Reviewer #1: Yes

Reviewer #2: Yes

4. Is the manuscript presented in an intelligible fashion and written in standard English?

Reviewer #1: Yes

Reviewer #2: No

5. Review Comments to the Author

Reviewer #1: This toned-down manuscript with a more accurate title is much improved. It might be helpful to have a reader fluent in English correct some of the minor grammatical and style errors, even though the narrative is easy to follow and is readily comprehensible. Overall, the paper sends a reassuring message to women confined to home during a pandemic.

Reviewer #2: This is a revised version of a manuscript in which the authors report various metrics related to late pregnancy, Caesarian deliveries and other aspects of pregnancy in the Wuhan, China region during the pandemic. The manuscript has been improved, and a large number of problems identified by the reviewers have been addressed.

There has been great interest in whether the infection of a pregnant mother would result in any changes in the fetus during late gestation or the infants after birth. Although the current study does not address this, the questions the authors do address have some value as well, but perhaps not enough to justify a publication in a broad and general interest journal like this. It seems like these types of data would be more appropriate for a specialty human journal that focused on Ob/Gyn, rather than a general interest journal like PLOS One.

There are still a number of problems with the English. It was recommended in the initial review that the authors consider using this type of service, or get some other editorial help from a native English speaker. Just a few of the types of problems that occur in the manuscript are listed below; this is by no means a complete list.

Line 114: you cannot start a sentence with a numeral in English.

Line 120: SCORES

Line 155: you cannot start a sentence with a numeral in English.

Line 197: change “pregnant women were trapped in their homes” to “pregnant women were confined to their homes”

Line 197-198: change “and it was not convenient for them to go out for taking a walk.” To “and their ability to exercise was limited.”

Tables 1, 3 and 4: The authors should review rules on significant digits. It does not seem correct to show the percentage values with two places to the right of the decimal. Would not whole percentages perhaps be better?

6. PLOS authors have the option to publish the peer review history of their article (what does this mean?). If published, this will include your full peer review and any attached files.

Reviewer #1: No

Reviewer #2: No

---

## [Author Response · Author response to Decision Letter 0]

30 Jun 2020

Response to Editor

Q1. This paper represents a resubmission of a previously submitted paper that was reviewed extensively. The authors were provided the opportunity to tone down their claims and ensure that throughout the paper it was clear that they are not examining how COVID-19 affects pregnancy outcomes or analyze such samples. They chose the former approach. The original reviewers have reviewed the revised paper and expressed conflicting recommendations. However, both agree that the work is not "blockbuster", but it may provide some comfort to pregnant women having to quarantine that it will not drastically affect their neonates. In reviewing the journal guidelines, it will allow for descriptive papers, such as this one. That being sad, it still needs to meet certain standards in terms of the English and previous and current reviewer concerns need to be addressed. It is recommended to have the revised manuscript proofed by an outside individual before it is resubmitted. Both reviewers have also provided helpful comments.

Response: Thanks for your comments. Following the reviewers’ comments, we have revised the manuscript carefully. In order to enhance the writing, a well-known outside professional editing institution has assisted us in improving the English writing of our manuscript with charges., which has substantially improved the quality of the paper.

Response to Comments of Reviewer 1 

Comments: This toned-down manuscript with a more accurate title is much improved. It might be helpful to have a reader fluent in English correct some of the minor grammatical and style errors, even though the narrative is easy to follow and is readily comprehensible. Overall, the paper sends a reassuring message to women confined to home during a pandemic.

Respond: Thanks for your positive comments with respect to our previous revision. In this round of revision, a professional institution has helped us in improving the English writing.

Response to Comments of Reviewer 2

Q1:There are still a number of problems with the English. It was recommended in the initial review that the authors consider using this type of service, or get some other editorial help from a native English speaker. Just a few of the types of problems that occur in the manuscript are listed below; this is by no means a complete list. 

Line 114: you cannot start a sentence with a numeral in English.

Line 120: SCORES

Line 155: you cannot start a sentence with a numeral in English.

Line 197: change “pregnant women were trapped in their homes” to “pregnant women were confined to their homes”

Line 197-198: change “and it was not convenient for them to go out for taking a walk.” To “and their ability to exercise was limited.”

Response: Thanks for your comments. In the revised manuscript, we have corrected all the mistakes you pointed out. In addition, a well-known professional institution has helped us in improving the overall English writing.

Q2. Tables 1, 3 and 4: The authors should review rules on significant digits. It does not seem correct to show the percentage values with two places to the right of the decimal. Would not whole percentages perhaps be better?

Respond:According to your comments, we have reviewed the papers with respect to retrospective study published on PLOS ONE, which have been listed below. We decide the follow their rules on significant digits and also the percentage values. The corresponding tables Tables 1, 3 and 4 have been revised carefully.

Chu S, Chen Q, Chen Y, et al. Cesarean section without medical indication and risk of childhood asthma, and attenuation by breastfeeding[J]. PLoS One, 2017, 12(9): e0184920.

---

## [Decision Letter · Decision Letter 1]

28 Jul 2020

Impact of Wuhan lockdown on the indications of cesarean delivery and newborn weights during the epidemic period of COVID-19

PONE-D-20-12899R1

Dear Dr. Zhao,

We’re pleased to inform you that your manuscript has been judged scientifically suitable for publication and will be formally accepted for publication once it meets all outstanding technical requirements.

Kind regards,

Cheryl S. Rosenfeld, DVM, PhD

Section Editor

PLOS ONE

Additional Editor Comments (optional):

Reviewers' comments:

Reviewer's Responses to Questions

**Comments to the Author**

1. If the authors have adequately addressed your comments raised in a previous round of review and you feel that this manuscript is now acceptable for publication, you may indicate that here to bypass the “Comments to the Author” section, enter your conflict of interest statement in the “Confidential to Editor” section, and submit your "Accept" recommendation.

Reviewer #1: All comments have been addressed

2. Is the manuscript technically sound, and do the data support the conclusions?

Reviewer #1: (No Response)

3. Has the statistical analysis been performed appropriately and rigorously? 

Reviewer #1: (No Response)

4. Have the authors made all data underlying the findings in their manuscript fully available?

Reviewer #1: (No Response)

5. Is the manuscript presented in an intelligible fashion and written in standard English?

Reviewer #1: (No Response)

6. Review Comments to the Author

Reviewer #1: The paper is now suitable for publication. The written English is much improved and the the conclusions suitably muted.

7. PLOS authors have the option to publish the peer review history of their article (what does this mean?). If published, this will include your full peer review and any attached files.

Reviewer #1: No

---

## [Editor Report · Acceptance letter]

3 Aug 2020

PONE-D-20-12899R1 

Impact of Wuhan lockdown on the indications of cesarean delivery and newborn weights during the epidemic period of COVID-19 

Dear Dr. Zhao:

I'm pleased to inform you that your manuscript has been deemed suitable for publication in PLOS ONE. Congratulations! Your manuscript is now with our production department. 

Kind regards, 

on behalf of

Dr. Cheryl S. Rosenfeld 

Section Editor

PLOS ONE